



# Seismic signature of the COVID-19 lockdown at the city-scale: A case study with low-cost seismometers in the city of Querétaro, Mexico.

Raphael S.M. De Plaen[1], Víctor Hugo Márquez-Ramírez[1], Xyoli Pérez-Campos[2], Francisco Ramón Zuñiga Davila-Madrid [1], Quetzalcoatl Rodríguez-Pérez[1], Juan Martín Gómez González[1], Lucia Capra[1]

[1]Centro de Geociencias, Universidad Nacional Autónoma de México, Juriquilla, Querétaro, Mexico.
[2]Instituto de Geofísica, Universidad Nacional Autónoma de México, Mexico City, Mexico.

*Corresponding author*: Raphael S.M. De Plaen (raphael.deplaen@geociencias.unam.mx)

## Abstract

Seismometers have detected the social response to lockdown measures implemented following the onset of the COVID-19

pandemic in cities around the world. This long-lasting lockdown has been a particular challenge in countries such as Mexico, where the informal economy constitutes most of the working population. This context motivated the monitoring of the mobility of populations throughout the various phases of lockdown measures, independently from people's access to the internet and mobile technologies.

Here we use the variation of anthropogenic seismic noise in the city of Querétaro (central Mexico) recorded by a network of

low-cost Raspberry Shake seismic stations to study the spatial and temporal variation of human activity in the city throughout the pandemic and during sportive events. The results emphasize the importance of densifying urban seismic networks and of tracking human activities without the privacy concerns associated with mobile technologies.

## 1   Introduction

In early 2020, the COVID-19 pandemic took the world by surprise and, given the absence of a treatment or vaccine it forced

the implementation of lockdown measures to prevent its spread. Like other countries in the world, Mexico progressively implemented confinement measures after the World Health Organization declared it a pandemic on 11 March (WHO, 2020). The first confinement measures, focused on closing schools and university campuses, were declared on 14 March and progressively evolved to a full suspension of non-essential activities on 24 March with the "Quedate en Casa" ("Stay at home") campaign (Secretaría de Salud, 2020a). Lockdown measures came to a slow, progressive relaxation from 1 June after

authorities declared the intention to gradually reopen the economy (Secretaría de Salud, 2020b).

A side-product of the lockdown and the subsequent limitation of non-essential activities was the significant reduction in seismic noise generated by human activities. This phenomenon, well documented at a global scale (e.g., Cannata et al., 2020; Dias et al., 2020; Lecocq et al., 2020; Poli et al., 2020; Xiao et al., 2020, p. 19), has yet to be analyzed in Mexico, which typically benefits from a high seismic station coverage as a result of its significant seismicity.





The response to lockdown measures in Mexico is of particular interest as it is challenged by the high level of informal economy in the country (~60%, *OECD Economic Surveys*, 2019). The uncertainty that transpires from this reality makes particularly important monitoring and characterizing how the lockdown measures were followed in the country.

Since the beginning of the crisis, community mobility data published by tech and social media companies contributed significantly to the analysis of social behavior changes following lockdown measures. However, these data result from

aggregates of the personal history of users' time spent at different categories of activities, which is likely to trigger privacy concerns, although it has undergone anonymization procedures (Aktay et al., 2020). In Mexico, such tools are also confronted with the limitations of access to the internet, especially creating a digital connectivity divide in rural areas, combined with the potentially limited utilization of smartphones.

Constraining these observations to urban areas where internet adoption and smartphone utilization were respectively estimated to 71.2% and 77.7%, as opposed to 39.2% and 53.8% in rural areas (Martínez-Domínguez & Mora-Rivera, 2020). Using geophysical sensing, such as fiber-optic distributed acoustic sensing (DAS), has the potential to mitigate these limitations and complement digital city sensing systems with direct measurements of public infrastructure usage (Lindsey et al., 2020).

Recently, anthropogenic seismic noise has also proved to be a good proxy for the reduction of human activities that followed

the 2020 lockdown measures worldwide, especially near urban areas (e.g., Cannata et al., 2020; Dias et al., 2020; Lecocq et al., 2020; Poli et al., 2020; Xiao et al., 2020). Anthropogenic seismic signals are generated by a wide range of sources ranging from traffic to sportive and cultural events (e.g., Diaz et al., 2020; Green & Bowers, 2008; Riahi & Gerstoft, 2015). When applied to the observation of the impact of confinement and social distancing measures, anthropogenic noise offers a perspective on the scale of the unprecedented measures authorities had to take to mitigate the spread of the pandemic. In most

cases in the literature, anthropogenic seismic noise amplitude correlates with community mobility data without the privacy-related concerns and limitations of using mobile technologies (e.g., Cannata et al., 2020; Dias et al., 2020; Lecocq et al., 2020).

Here, we focus on the city of Querétaro, 200 km northwest of Mexico City. Even though the area is not affected by frequent strong local seismicity by Mexican standards, it is currently equipped with a network of seven Raspberry Shake 4D seismic stations spread throughout the city. The relative density of the network combined with the low local seismicity makes this city

an ideal laboratory to explore the impact the lockdown measures on seismic noise and the general signature of human activities observed by seismic instruments.

The lockdown in Mexico was structured in three phases with growing constraints as the number of cases increased in the country from mid-March until June (Table 1). From June on, the authorities implemented a plan towards *la nueva normalidad* (the new normality, NN) with a monitoring system regulating the use of public spaces according to the risk of contagion of

COVID-19. This system assigns a stoplight color to each state ranging from red to green as a function of the ongoing epidemiological risk (Secretaría de Salud, 2020, Semáforo COVID-19, Table 2). The different phases of lockdown and colors of stoplight are associated with critically additional constraints on non-essential activities. Under the assumption that most people respected those regulations, the anthropogenic noise is expected to vary accordingly.





Table 1 Descriptions of the early phases of the federal lockdown in Mexico (Secretaría de Salud, 2020a).

| Phases | Announcement/ implementation dates | Description |
|---|---|---|
| Phase 1 | 14-03-2020/21-03-2020 | Importation of the virus with dozens of identified infected travelers entering the country. Closing of schools and universities with early recess. |
| Phase 2 | 24-03-2020 | Community propagation of the virus with hundreds of infected people and the origin of the infections is no longer well identified. Suspension of classes, events and meetings of more than 100 people. Suspension of work activities that involve the mobilization of people in all sectors of the society and intensification of basic sanitary prevention measures. Start of the "Quédate en Casa" campaign. |
| State of Sanitary Emergency | 30-03-2020 | Immediate suspension of non-essential activities in the public, private and social sectors; |
| Phase 3 | 21-04-2020 | National epidemiological stage with thousands of confirmed infectious cases throughout the country. The measures to stay in homes are reinforced with the limitation of non-essential activities. |


Table 2. Categories of the epidemiological risk stoplight during the "new normality" in Mexico (Secretaría de Salud, 2020b).

| Color | Description |
|---|---|
| Red | Only essential businesses are allowed, people are invited to shelter at home. |
| Orange | Essential businesses are allowed, non-essential businesses are allowed to operate with up to 30% of the personnel, open public spaces are opened with a limited capacity. |
| Yellow | All businesses are allowed to operate. Open public spaces are open at regular capacity, and closed public spaces at reduced capacity. All activities must be carried out with basic preventive measures. |
| Green | All activities are allowed at regular capacity, including school. |





## 2    Material and method

### 2.1    Data

The Querétaro metropolitan zone (QMZ) is monitored by a seismic network operated by the Center of Geosciences (UNAM-Campus Juriquilla). As of 2020, the network, designed for studies of local and regional events, is composed of seven Raspberry Shake 4D stations that each incorporate a vertical component geophone and a 3-component accelerometer (Figure 1).

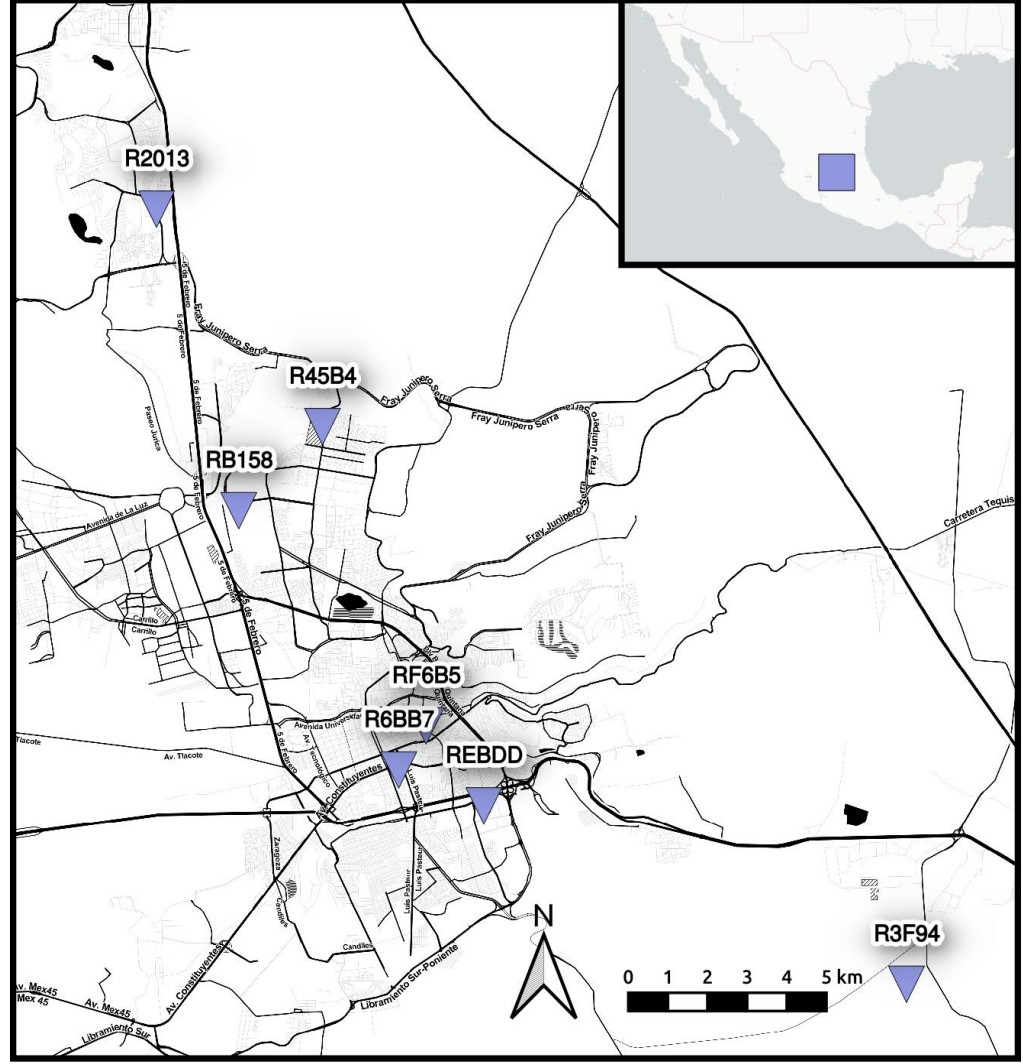

Figure 1. The seismic network of the metropolitan zone of Querétaro. The inverted triangles are the location of the stations used in this study. The square in the inset shows the location of the city of Querétaro (map tiles by Stamen Design, under CC BY 3.0, data by ©OpenStreetMap contributors 2020 distributed under a Creative Commons BY-SA License).





These low-cost sensors have proved to efficiently densify backbone networks to monitor the vertical motion of local and global
     seismic activity from all magnitudes of earthquakes and remain on scale also to measure the motion generated by larger more
     powerful local earthquakes (Anthony et al., 2019). All stations record continuous data at a sampling rate of 100 Hz except for
     station R2013 recording at 50 Hz. Of the three stations installed near the historic center, RF6B5 and R6BB7 are both installed
     in cultural centers that suspended their activities since the beginning of the confinement, and REBDD is installed in the
Corregidora soccer arena, which has a capacity of ~34000.

     The stations located further from the center of the city are installed in university campi (R2013, R45B4 and R3F94), and in
     the Benito Juarez industrial park (RB158). All the network stations are within 5 kilometers of four-lanes highways with 90-
     km/h speed limits and significant traffic. The period analyzed here extends from 1 November 2019 to 1 October 2020.

## 2.2   Anthropogenic noise analysis

85   The QMZ network was designed to study natural processes, specifically local and regional tectonic earthquakes, rather than
     human-generated signals. Nevertheless, the recorded signal is subjected to a significant amount of noise, whether systematic
     (instrumental noise, numerical conversion), natural (first and second microseisms, atmospheric, etc.), or human-made. The
     latter includes  a broad spectrum of potential sources typically recorded between 2 and 20 Hz, such as traffic, public
     transportation, pedestrians, and industries.(e.g., Diaz et al., 2020; Poli et al., 2020).

90   Human-generated noise is also characterized by a strong diurnal variability and higher amplitudes during business days than
     weekends and public holidays. As a result, we measured the anthropogenic noise by using the signal between 7 am and 7 pm
     local time. Using the same method as Lecocq et al. (2020), we computed the power spectral density (PSD) using Welch's
     method (Welch, 1967) from 30-minute windows with a 50 percent overlap.

     The time series were extracted from the root mean square (seismic RMS) amplitude of the seismic signal. The seismic RMS
95   was first implemented using a narrow band of 1 Hz for multiple frequencies ranging from 1 to 20 Hz to better identify the
     stable frequency range most affected by the lockdown measures (Figure 2). The resulting time series were then used to identify
     the Spearman correlation coefficient between each narrowband seismic RMS and each community mobility category
     (Figure 3).







Figure 2. Variations of seismic RMS measured with narrow 1 Hz frequency band from 1 to 20 Hz at all stations. The typical succession of business days and weekends can clearly be observed, but the largest reductions in seismic noise levels are identified during the Christmas holiday (green dashed lines are Christmas and New Year, respectively) and following the implementation of lockdown measures mid-March 2020 (white dashed line).





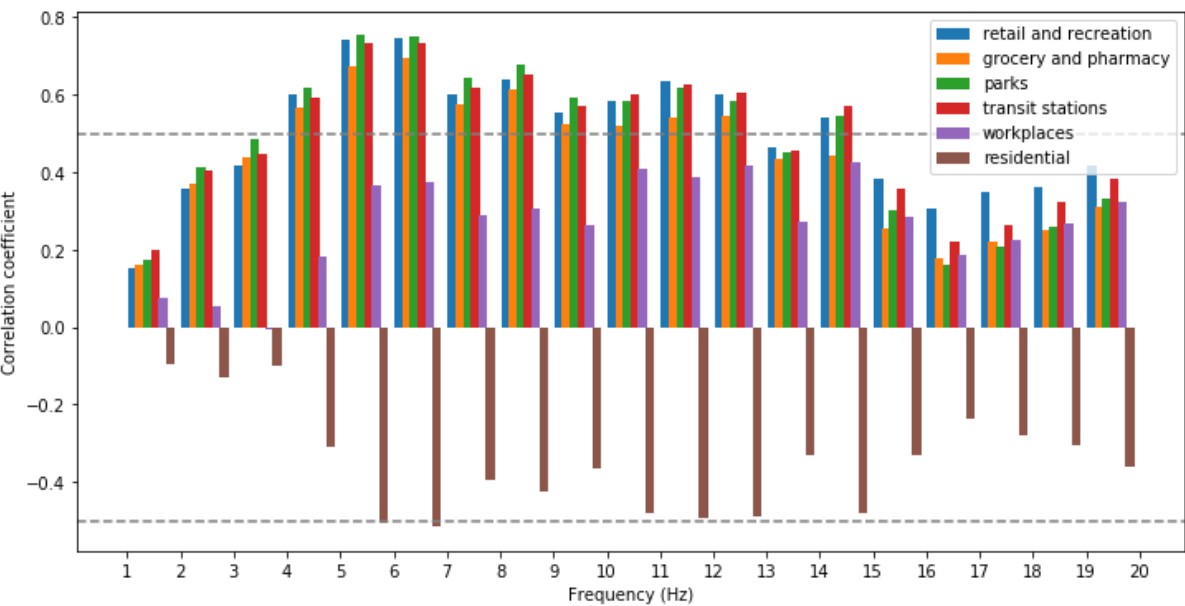

Figure 3. Correlation coefficient between the seismic RMS in narrow 1 Hz frequency band from 1 to 20 Hz at station R6BB7 and the Google community mobility in the high categories of retail and recreation, groceries and pharmacy, parks, transit stations, workplaces, and residential premises. The highest correlation and anticorrelation are identified between 4 and 14 Hz.

## 2.3 Mobility data

Since the beginning of the pandemic, Google, along with other tech and social media companies (e.g., WAZE, Apple, Facebook), started sharing community mobility data created with aggregated, anonymized sets of data from users' location history (Google, 2020). Here we focus on data provided by Google as they are likely to sample a larger portion of the population and more heterogeneous social backgrounds (González Alma Delia et al., 2018). These reports offer analyses of movement trends over time by geography, across different high categories of places defined as retail and recreation, groceries and pharmacies, parks, transit stations, workplaces, and residential. The data reflects the time visitors spent in the different categories of places compared to baseline days, calculated as the median value from the arbitrary 5-week period from January 3 to Feb 6 2020. Each value in the report is, therefore, a measurement relative to 7 individual values of the corresponding day of the week during the baseline period.

These data are imperfect on many levels, especially when applied to a city in Mexico. The raw data only samples the portion of the population with stable access to the internet and actively using the Google services. Evidence shows that the urban population is far better connected than the rural, and Android is by far the leading operative system with over 80% of the Mexican market (González Alma Delia et al., 2018; Martínez-Domínguez & Mora-Rivera, 2020).





The data are anonymized using libraries of differentially private algorithms, which add Laplace noise to protect each metric with differential privacy (Aktay et al., 2020). Furthermore, metrics for which the geographic region is smaller than three

kilometers squared, or for which the differentially private count of contributing users after noise addition is smaller than 100, are discarded. The original data is by design far from transparent, leading to significant uncertainties on features such as the number of people, or the delimitation of the locations sampled (e.g., Lecocq et al., 2020).

In this work, we used data for the state of Querétaro, the smallest spatial granularity level available for the region. Since the QMZ contains ~64% of the population and the largest portion of the urban population likely to use such mobile technology in

the state (INEGI, 2016), we consider this approximation acceptable for our purposes.

With those limitations, in the absence of a better option, these community mobility data offer the best solution to specifically track the social response of the population to lockdown measures and compare them to ambient noise monitoring.

## 3 Results and discussion

Significant reduction of the seismic RMS following the implementation of lockdown measures are observed at all narrow

frequency bands between 1 and 20 Hz. Higher correlation coefficients between the community mobility and the Seismic RMS were obtained between 4 and 14 Hz, leading us to concentrate on that frequency range.

To better interpret the variation of seismic RMS relative to the community mobility, we converted those values to percentage of change according to the corresponding day of the week during a baseline period. The baseline is calculated as the median value for the corresponding day of the week from 13 January to 2 March, excluding 3 February, which was a holiday. This

baseline period is different from the one used for the Google community mobility; it is a larger period, starting later to avoid any influence from the end of the Christmas and New Year school holidays. The baseline period finishes before the announcement by the World Health Organization that COVID-19 had officially become a pandemic.

The resulting time series show less impact from weekends, but they are characterized by lower noise levels (Figure 4). Instead, days such as public holidays with anthropogenic noise levels that are out of the ordinary are specifically emphasized with

reductions from -20 to -50% registered at all stations. With the implementation of the first lockdown measures, the relative noise level started to progressively decrease at all stations except for RB158, located in the Benito Juarez industrial park, where the noise level remained 20 to 50% higher than the baseline from early March. Noise levels at this station decreased after the declaration of sanitary emergency and on the Good Friday holiday but then displayed a slow progressive increase until June 1st and significantly remained above baseline. This suggests a sustained higher than normal level of activity in the close

vicinity of the industrial park even as the Mexican government suspended non-essential activities in the public, private, and social sectors. This high level of activity could be the result of essential activity such as the delivery of food and supplies and would explain why this higher level of activity was later sustained.



Figure 4. Variation of seismic RMS in percent at all 7 stations of the network relative to the median of the corresponding day of the week during the baseline period of 13 January to 2 March.(A). The public holidays are indicated with blue dashed lines and the implementation of lockdown measures is indicated as the red dashed line. The purple span corresponds to the early lockdown phases and the red and orange spans after 1 June are the corresponding stoplight color for the "New Normality". (B) to (D) are the maps of those corresponding variations on the 15 February, 25 March, and 15 May 2020, respectively (Map tiles by Esri, under CC BY 4.0, data by ©OpenStreetMap contributors 2020 distributed under a Creative Commons BY-SA License).

All the other stations displayed significant noise reduction until at least the end of the Easter Holidays. During this period, station R45B4, located at the Universidad Tecnológica de Querétaro Campus (UTQ), displayed the lowest reduction with noise levels of less than 10%, followed by sustained increased levels up to 30% above the baseline as a result of roadworks on a





neighboring avenue. This high level of seismic RMS at this station holds up until the end of August. It can safely be attributed
to the roadwork in the nearfield rather than a specifically lockdown-related human activity change.

Figure 5. (A) Median of the variation of the seismic RMS at all stations, relative to the median of the corresponding day of the
week during the baseline period of 13 January to 2 March, and variation of the Google community mobility in 6 high-level
categories. The public holidays are indicated with blue dashed lines and the implementation of lockdown measures is indicated
as the black dashed line. The purple span corresponds to the early lockdown phases and the red and orange spans after 1 June
are the corresponding stoplight color for the "New Normality". Cross-plot between the median seismic RMS of the network
and the Google community mobility in 6 high-level categories during the early lockdown phases (B) and including the new
normality (C). The spearman correlation coefficients between the median seismic RMS and each category of community
mobility and the corresponding p-values are provided in the legend.






The median of all the stations better illustrates this change, with a progressive decrease through the lockdown phases 1 and 2 and the state of sanitary emergency until the Good Friday holiday that reached ~40% under the baseline (Figure 5A). The median then slightly increased followed by a decrease after phase 3 of the lockdown up until May 1st (Labor Day holiday). The noise then increased again progressively until the implementation of the stoplight system of the NN on June 1st. From the end of the Easter holidays to June 1st, stations REBDD, RF6B5, R3F94, R2013, and R6BB7 maintained an average noise level of -19.31, -27.93, -31.99, -32.70, and -35.68% respectively.

During the NN, the stoplight changed from red to orange with a subsequent relaxation of restrictions accompanied by a consistent progressive increase of the level of noise. As the infection risk increased, the stoplight was turned red again on 20 July for two weeks, during which the seismic RMS slightly decreased again. The stoplight was then turned orange again until 1 October, when it changed to yellow, by which the seismic RMS increased back to baseline levels.

From mid-August, station R2013 at UNAM campus displayed a significant increase in seismic RMS which held up on average 30% above the baseline. This sharp increase when some restrictions were still in place resulted from the relocation of the instrument from an office to a noisier site.

### 3.1 Sportive events

The station REBDD located in the Corregidora soccer arena also offers an interesting perspective on the contribution of the public attendance to the recorded seismic noise. Seismometers in soccer stadiums are regularly used in Mexico as an entertaining tool to teach basic scientific research methodologies and seismology to undergraduate students (e.g., Melgar & Pérez-Campos, 2011). In Querétaro, all games had been canceled with the implementation of the early lockdown measures until mid-July, when authorities allowed games without public attendance. This situation makes for a good opportunity to compare observations during games played before and after the beginning of the confinement. In what follows, we look at data from two games played by the first division Gallos Blancos of Querétaro soccer club against the Cruz Azul soccer club of Mexico City (3 August 2019 and 12 August 2020, Figure 6) as well as one against the Águilas del América soccer club of Mexico City (9 February 2020 and 16 August 2020, Figure 7). The seismic RMS is calculated with the method described above but with a window reduced to 100 seconds.

Both games played against the Cruz Azul team were won by Gallos Blancos 3-0 and 1-0, respectively. During the first game, which occurred with an audience of 29,339 supporters, a clear signature can be identified at the moments Gallos Blancos scored goals with a significant increase (Figure 6) in noise. The half-time and especially the end of the game are also characterized by a relatively higher noise level than the rest of the game, which could potentially translate in the enthusiasm of the supporters at the final score. This development contrasts with the August 16th 2020 game played without an audience. The entire game remained at a relatively low level of noise very close to the median of the week for that time of the day (Figure 6). The only goal, at the 29th minute of the game, does not emerge from the background noise as only the players were present to celebrate it.



Figure 6. Seismic signal recorded by station REBDD during the football games between the Gallos Blancos of Querétaro and
210 the Cruz Azul of Mexico City in the Corregidora Soccer Arena on 3 August 2019 and 12 August 2020, respectively. (A) and
(B) are the waveforms, and (C) and (D) the spectrogram for each respective game. (E) compares the seismic RMS between
the two games.





Figure 7. Seismic signal recorded by station REBDD during the football games between the Gallos Blancos of Querétaro and the Club América of Mexico City in the Corregidora Soccer Arena on 9 February and 16 July 2020, respectively. (A) and (B) are the waveforms, and (C) and (D) the spectrogram for each respective game. (E) compares the seismic RMS between the two games.

A similar pattern is observed during the matches against the Águilas del América team. The first game on February 9th 2020, with 34,050 supporters, was won 1-2 by América with noise levels consistently higher than the baseline for that time of the day on a Sunday and with clear spikes every time a goal was scored by either team. This pattern again shows the impact of the





supporters on the recorded seismic noise, which is further emphasized in contrast to the July 16th 2020 game where goals scored do not significantly emerge from the background signal.

## 3.2    Correlation with mobility

The different categories of community mobility were compared to the variations of noise levels to identify the one most affected by lockdown measures. The different correlation coefficient obtained for each category of community mobility sheds light on the impact of lockdown measures (Figure 5). The metrics of the community mobility are based on the data of Google users in the state of Querétaro, who use Location History, the majority of which are assumed to live in the QMZ. The median of network stations rather than individual stations was therefore compared to the community mobility to better correspond to the sampled population.

The median seismic RMS has a high correlation with all the categories except the residential one but retail and recreation (0.61), transit stations (0.58), and workplace (0.57) categories are the highest. With non-essential workers invited to shelter at home, this is an expected outcome reinforced by the strongly anticorrelated (-0.62) residential category (Figure 5C). None of the stations are installed in an overwhelmingly residential area, mobility in that category is therefore not expected to be c losely mapped by seismic noise, hence the anticorrelation. The relationship between seismic RMS and the different categories of community mobility noticeably higher during the early phases of lockdown (Figure 5B). This could reflect the more complex mobility pattern people adopted during the new normality, as the authorities progressively encouraged the reopening of the economy with the help of sanitary measures.

Anthropogenic seismic noise recorded in cities is typically generated by various processes suc h as our cultural and sportive events, individual and public commuter traffic and the production as well as transportation of different kinds of goods (e.g., Groos & Ritter, 2009). Traffic especially heavily influences the frequency range from ∼1 Hz to more than 45 Hz with variation with respect to urban location due to the influence of subsurface conditions.

Surface waves generated by vehicles in stratified soil has been shown to be mainly restricted to 2–20 Hz the frequency band. The vehicle-induced ground vibrations are generated by the pitch, and axle hop modes of the vehicle coupling to the road. Although far-field vibration levels depend on vehicle speed, road unevenness, vehicle, and soil characteristics, vibration frequencies attributed to axle hop modes depend on vehicle distance and velocity (Lombaert et al., 2000; Lombaert & Degrande, 2001).

The seismic noise recorded by stations all within 5 km of the main roads in the state is likely a good proxy for the variation of traffic volume. With a lot of people working from home or losing their jobs during the lockdown, our results could in large part indirectly show the decrease in commuter traffic.



## 4    Conclusions

Like in a lot of populated places around the world, the effect of lockdown measures was observed as a decrease of high-frequency seismic noise in the city of Querétaro. Our results specifically show the benefit of using a network of low-cost

Raspberry Shake stations to investigate the temporal and spatial variation of the anthropogenic noise at a city-scale.

While the tracked variation of human-generated noise has a high similarity with community mobility report obtained from mobile technologies, it yields a finer resolution that could effectively single out localized zones of increased noise level caused by more activity, such as the industrial park Benito Juarez, or construction work, such as at the campus of the Universidad Tecnológica de Querétaro.

The privacy concerns associated with mobile technology-based community mobility data require a low resolution to anonymize original user data. Along with a higher resolution, anthropogenic seismic noise has the advantage of being anonymized by definition and does not require the population to trustfully or unwillingly share their personal data.

While we observed the contribution to anthropogenic noise, such as construction work and sportive events, at a very local scale, traffic appeared to dominate observation at the city scale. While observing variations of traffic volumes in the seismic

signal is not monitoring the lockdown per se, it proves to be a reliable proxy for human activity, especially at the city-scale with such a network.

### Acknowledgments

This work is supported by the grant CONACYT-299766. R.D.P. acknowledges support from the UNAM-DGAPA postdoctoral scholarship.

**Data availability**

Data for stations R3F94, RB158 and R3F94are openly accessible as part of the citizen scientist earthquake monitoring network (AM) through FDSN Web Services (https://doi.org/10.7914/SN/AM). The remaining data used in this study is available upon request to the corresponding author.

### Author contributions

RDP, VHM and XPC. designed the study. All the Authors analysed the seismic data. RDP, VHM and XPC  wrote the paper. All the authors interpreted the results and revised the article.

### Competing interests

The authors declare that they have no conflict of interest.



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
