# Peer review of "Seismic signature of the COVID-19 lockdown at the city-scale: A case study with low-cost seismometers in the city of Querétaro, Mexico."

_Solid Earth, 2020_

## Referee Comment (RC1) · Anonymous Referee #1 · 21 Dec 2020

This manuscript uses ground motion data from 7 Raspberry Shake RS4D stations located throughout Queretaro, Mexico to demonstrate reductions in seismic noise in response to COVID19 lockdown measures implemented by the Mexican government to mitigate the spread of the disease. The paper then compares these reductions with independently attained mobility data from google and shows the expected positive correlations between seismic noise and transportation, shopping, and recreation. Additionally, seismic recordings are compared at two soccer games between identical clubs both prior to the pandemic (with approximately 30,000 spectators) and after the

pandemic when in-person attendance was forbidden. Processing of the seismic data is carried out using the Lecocq et al (2020) software package and is appropriate for the study. Overall, I found the study convincingly demonstrated reductions in cultural noise in response to lockdown measures. I enthusiastically recommend the manuscript for publication in the EGU special issue on Social Seismology and the effect of COVID19 lockdowns following moderate revisions to substantially improve the presentation and reproducibility of the work.

Despite some of these presentation issues, I found the paper fun to read – something that can't often be said about technical manuscripts! Nice work on the study!

Main Points

I think one of the conclusions of the study, "that traffic noise is dominating the high-frequency noise environment within cities" could use more support in the discussion section. It would be good to have a better tie-in between the observations in this study with previous work to demonstrate this conclusion.

Additionally, I think that several (mostly tangential) points in the manuscript are misleading or were stated incorrectly. These should be addressed in a revision and I point them out in the line edits below.

Otherwise, I have no overarching concerns from a scientific standpoint. The seismic and mobility data was analyzed using known, accepted methods by the seismological community that is appropriate for the study. Overall, Figures are of high-quality and are easy to parse, though some more information in the caption would be useful.

The following line edits are my opinion and are meant to improve the presentation and reproducibility of the manuscript.

Line Edits

L10: Long-lasting "pandemic" for "lockdown" – this is being discussed in a global context, lockdown measures were highly variable across the globe – lumping them all

together and calling them "long-lasting" when some cities didn't even implement measures seems inaccurate to me.

L16 and Elsewhere: "sporting" for "sportive." In this manuscript "Sportive Events" refers to soccer matches, not an endurance bike race (the typical definition of a "Sportive event"

L27: Suggest discussing more broadly here the changes in society that have been observed as resulting from COVID19 lockdown measures, and then drilling down into the seismological observations later. Right now, this seems redundant with L49-50.

L30-34: I found these sentences very difficult to follow. Suggest re-writing and including a definition of "informal economy" as seismologists and geophysicists probably don't know what this is.

L40 – This sentence seems like it should be included in the previous paragraph, and as written doesn't realty make sense. I would re-write it as: "For instance, internet adoption in rural areas of Mexico is estimated at 39.2% in urban areas with 53.8% of the population owning smartphones. In contract, 71.2% of urban residents have access to the internet and 77.7% use smartphones."

L41 – Bringing up DAS technology to monitor ground motions seems totally out of place here (seismic analysis hasn't even been introduced yet). This would be better to bring into the discussion.

L49 – Another sentence that makes no sense as written. Suggest splitting into 2 sentences

Suggested wording ". . ...correlates with mobility data (references). Therefore, analysis of seismic noise may off the ability to attain mobility information without the privacy-related concerns and limitations of using mobile technology."

L52: "Even though this area has lower seismicity rates than other parts of Mexico, it is. . .. . .."

[Figure]

Table 1: Dates and events for "phase 1" are confusing because 2 dates are given. Can these be separate rows in the table or is the 14-03-2020 event needed?

Table 2: Red row: suggest "encouraged" for "invited" – "invited to shelter at home" isn't really correct English.

Figure 1 – Suggest a box outlining the "historic center" as stated on L78

L75 – Anthony et al, did not use Raspberry Shakes to densify networks as this sentence suggests - rather they tested the performance of the sensors and found them to be acceptable for use u regional networks. "from all magnitudes" is misleading – for instance, a Raspberry Shake cannot see a M5 teleseism. Finally, it might be nice to note that only data from the geophones is considered.

Suggested edit: "These low-cost sensors were demonstrated to perform suitably well for monitoring a large range of local and regional earthquake magnitudes (Anthony et al., 2019). However, while the accelerometers are capable or recording larger, more powerful earthquakes, they have high self-noise levels and are not able to resolve most cultural activity. Therefore, we restrict our analysis in this study to only the vertical component geophone data."

L81: Are these multiple universities? If so, it would be good to specify which Universities are included here.

L87 – remove "numerical conversion" – I'm not aware of this noise source unless it is digitizer noise, which I would think would fall under the "instrumental noise" umbrella. L87 – Just leave it as "microseisms" – these are called "primary" and secondary microseisms (not first and second), but you won't be able to observe the primary with a Raspberry Shake.

L93 – Would probably be good to specify the amount of smoothing used in the Lecocq et al processing algorithm, especially since the data is then bandbassed into 1 Hz bins, and depending on smoothing in this step, these frequencies may not actually be being

isolated.

L94 – Would be good to make it clear that these are displacement RMS timeseries data (As opposed to acceleration or velocity).

L95 – Would be good to specify how the 1 Hz RMS displacement bins were constructed. Was the data simply bandpassed? How many poles on the bandpass filter and where were they?

Figure 2 – Nice Figure! I would specify in the caption that what is being plotted is something along the lines of "normalized RMS displacement in 1 Hz frequency bins." I also suggest noting the move of R2013 directly in the Figure since the change of noise is so striking.

Figure 3 – It would also be good to specify in the caption the time-period over which the correlation analysis was performed as well as the meaning of the dashed lines (I assume they indicate the threshold for correlations to be significant at the 95% Confidence Interval?)

L116 – Can the normalization time period of the mobility data be changed to match that of the seismic (L138)?

L135 - Probably worth noting in the text why station R6BB7 was chosen for this analysis as well as that the determined 4-14 Hz frequency band is identical to that used in the Lecocq et al. (2020) study.

L152 – Is it possible that the work in the industrial park is seasonal? E.g. there is more activity in the spring and summer? It might be worthwhile to look at another year of data to make sure that seasonal trends are not being interpreted as COVID19-related here.

Figure 4 I don't think panels b-d were discussed in the text. Also, it wasn't clear to me if only a single day of data was being plotted or a date range. I suggest removing them from the manuscript if they are not needed. Figure 4A stands nicely on its own I

think. . ..

It would be good to specify in the caption that this is 4-14 Hz noise (at least I think it is).

What do the red and green shaded areas below the RMS noise curves mean?

The "blue" holiday lines look "green" to me in the Figure.

It might be helpful to plot all stations on the same Y-axis to facilitate direct comparison between stations. It took me a while to realize the axis was not consistent.

L160 – Date of Easter would be helpful

L161- ". . ...displayed the smallest change with noise levels dropping by less than 10%,. . .."

Figure 5 – Consider adding a dashed "0-line" to the plot as in Figure 4a.

The holiday dashed lines look more black than blue to me. . .especially early on. It might be better to use solid and dashed lines here to distinguish between lockdown measures and holidays.

Give date range of cross plots in Figures 5b and 5c. It wasn't clear to me over which time period these correlations were performed – particularly for Figure 5b

L177 – Suggest giving the data of "Good Friday Holiday"

L181 – Suggest referencing figure 4a here

L190 – citations of Vidale (2011; SRL) and Diaz et al (2017, Scientific Reports) are probably warranted here as they also considered the seismic signatures of fans responding to scoring events at football games (Vidale is American football, but same idea).

L204 – This conclusion that the uptick in noise after the game is related to the home-team winning seems to conflct with Figure 7, where the same uptick is seen in 2019 even though Gallos Blancos lost the game. . ...Could this uptick not be due to people

starting their cars and driving home after the game?

Figures 6 and 7

- Label colorbar - Specify the frequency band plotted in the "E" panels. - Add median noise level for the week (as discussed in L205) to panel E

L232 – Over what time period were these correlation coefficients attained from? They don't match Figures 5b or 5c. . . . . . I assume it is from a longer period than either of these plots?

L234 – This sentence seems to argue that the only reason a positive correlation between seismic noise and people staying at home isn't observed is because the seismometers were not installed near residential areas. I don't think this is correct.

I think the negative correlation with "residential areas" is expected as people don't make many seismic signals when they are sitting at home watching TV or working remotely. As is alluded to in the conclusion – most high frequency noise is likely coming from traffic – and this is simply reduced when people are staying at home.

L242 – It seems like the argument is being made here that traffic noise is the dominant high-frequency noise source that is being modulated throughout the study? Can this be made more clear? Can you find some studies that show this? This is stated as a conclusion on L264, so a bit more support and clear thought that traffic is indeed the dominant noise source is warranted here.

---

## Referee Comment (RC2) · Anonymous Referee #2 · 5 Jan 2021

Review of SE_2020_194: Plaen et al., "Seismic signature... Querétaro, Mexico"

Several papers have now been published describing the reduction of seismic ambient noise due to Covid-19 lockdown measures. So, this is not really an entirely new contribution. However, high frequency ambient seismic noise recorded in cities has many different contributing sources, mainly related to traffic, and it is not always easy to separate their sources and estimate their distance range. In this respect, the Querétaro case study, describing different noise behaviors in different parts of the town, is interesting and deserves publication as it may contribute to further more detailed analyses.

[Figure]

The paper is generally well written and clear, and the figures are of good quality. Some points, however, would benefit from better descriptions. I recommend publication with minor revisions, especially addressing the main points below.

Main points:

1) Mobility measures. It would be good to describe more clearly the meaning of the "Google mobility index". For example, does a 30% increase in residential mobility mean that there are 30% more people (i.e., 30% more cell phones) in residential areas, compared to the baseline period? Does the Google "mobility index" record only when people change their location ("movement trend" as in lines 113-114)? Or when they use their location app in the cell phones, even if not moving ("the time spent" in each category of place, as in lines 115-116). This is important to interpret the correlations of Fig. 5, for example.

2) In line 143, the sentence "The resulting time series show less impact from weekends, but they are characterized by lower noise levels ( Figure 4)" was not too clear for me. You mean that noise on weekends also decreased compared to the baseline, but the decrease was not as large as for weekdays?

3) Lines 151-152 explains the increase in noise levels in the industrial area as possibly due to increase of "delivery of food and supplies". Is this just a hypothesis? Is there a way to help confirm this explanation?

4) section 3.2 Correlation of noise with mobility:

4.1) Lines 235-236. I did not quite understand the explanation for the anti-correlation with mobility in the residential areas. if increased mobility in the residential category indicates more people are staying at home, then less traffic will occur both in the residential areas as well as elsewhere in the city.

4.2) Lines 236-239. I did not understand why the lower correlations in the "New norm" period, compared with the "Early lockdown phase" imply that the mobility pattern is

"more complex" during the relaxation period. For example, a 30% mobility reduction (assuming this is mainly related to traffic reduction) should produce the same RMS noise reduction, independent of the lockdown or new-norm period. I do not understand why traffic (the main source of seismic noise) should relate to the google mobility in different ways in the lockdown and in the new-norm phases.

The problem, I think, is that correlation coefficients may not the best measure as they are much influenced by the scatter. The slope of the straight-lines in Fig. 5 may be a better proxy for the relationship between mobility and seismic noise. For example, the red lines (transit stations) have very similar slope in both phases, during the lockdown (Fig. 5b) and during the whole period (Fig. 5c): about 37% noise reduction/100% mobility reduction.

5) In section 4 (Conclusions) the authors seem to conclude that "traffic appeared to dominate observations" of seismic noise. Strictly speaking, the paper does not "prove" this link between traffic and seismic noise. However, all stations are near roads and highways with heavy traffic ("< 5 km"); in addition, it is well known from the literature that traffic is the main contributor to seismic noise in stations close or within cities. So, it would be better to conclude that the observed correlation between seismic noise and mobility is consistent with the traffic-dominated nature of seismic noise in urban areas.

Minor points:

a) Fig. 1: Please edit the position of label R6BB7 as it hides one of the other stations.

b) Fig. 4: Please indicate the "baseline period" (perhaps along the time axis) as it may help the readers understand the plot more easily.

c) Fig. 5: I suggest to remove the probability level, so as not to clutter the figure, leaving only the r-value. This will also avoid hiding some of the data points. All probabilities are extremely small (because the number of points is very large) - it would be sufficient to mention in the text that all correlation coefficients are highly significant (probabilities

less than 10E-15).

d) Section 3.1 Sportive events: The analyses of the two football matches is interesting, but it does not add much. It only says that the noise from 22 players running in the field is insignificant compared with that of 30000 supporters jumping in the stadium.

e) Paragraph 230: the correlation coefficients mentioned in this paragraph are different from the ones in Fig. 5. Please check.

---

## Author Comment (AC1) · 14 Jan 2021

This manuscript uses ground motion data from 7 Raspberry Shake RS4D stations located throughout Queretaro, Mexico to demonstrate reductions in seismic noise in response to COVID19 lockdown measures implemented by the Mexican government to mitigate the spread of the disease. The paper then compares these reductions with independently attained mobility data from google and shows the expected posi-

tive correlations between seismic noise and transportation, shopping, and recreation. Additionally, seismic recordings are compared at two soccer games between identical clubs both prior to the pandemic (with approximately 30,000 spectators) and after the pandemic when in-person attendance was forbidden. Processing of the seismic data is carried out using the Lecocq et al (2020) software package and is appropriate for the study. Overall, I found the study convincingly demonstrated reductions in cultural noise in response to lockdown measures. I enthusiastically recommend the manuscript for publication in the EGU special issue on Social Seismology and the effect of COVID19 lockdowns following moderate revisions to substantially improve the presentation and reproducibility of the work. Despite some of these presentation issues, I found the paper fun to read – something that can't often be said about technical manuscripts! Nice work on the study!

>We thank the reviewer for the positive comments and useful suggestions.

Main Points I think one of the conclusions of the study, "that traffic noise is dominating the high frequency noise environment within cities" could use more support in the discussion section. It would be good to have a better tie-in between the observations in this study with previous work to demonstrate this conclusion. Additionally, I think that several (mostly tangential) points in the manuscript are misleading or were stated incorrectly. These should be addressed in a revision and I point them out in the line edits below. Otherwise, I have no overarching concerns from a scientific standpoint. The seismic and mobility data was analyzed using known, accepted methods by the seismological community that is appropriate for the study. Overall, Figures are of high-quality and are easy to parse, though some more information in the caption would be useful. The following line edits are my opinion and are meant to improve the presentation and reproducibility of the manuscript.

Line Edits L10: Long-lasting "pandemic" for "lockdown" – this is being discussed in a global context, lockdown measures were highly variable across the globe – lumping them all together and calling them "long-lasting" when some cities didn't even imple-
ment measures seems inaccurate to me.

>Changed as suggested.

L16 and Elsewhere: "sporting" for "sportive." In this manuscript "Sportive Events" refers to soccer matches, not an endurance bike race (the typical definition of a "Sportive event"

>Changed as suggested throughout the manuscript.

L27: Suggest discussing more broadly here the changes in society that have been observed as resulting from COVID19 lockdown measures, and then drilling down into the seismological observations later. Right now, this seems redundant with L49-50.

>We removed the seismic noise sentences from L27 and focused those observations around L49 as suggested. It now reads as follows: "Recently, anthropogenic seismic noise has also proved to be a good proxy for the reduction of human activities that followed the 2020 lockdown measures worldwide, especially near urban areas (e.g., Cannata et al., 2020; Dias et al., 2020; Lecocq et al., 2020; Poli et al., 2020; Xiao et al., 2020). This phenomenon, well documented at a global scale has yet to be analyzed in Mexico, which typically benefits from a high seismic station coverage as a result of its significant seismicity. "

L30-34: I found these sentences very difficult to follow. Suggest re-writing and including a definition of "informal economy" as seismologists and geophysicists probably don't know what this is.

>We added the following explanation: Activities of the informal economy, which exclude illicit ones, are not taxed, regulated or monitored by any form of government and as a result not subject to social protection or other types of employment benefits (OECD/ILO 2019).Workers and economic units participating in this economy do not have access to secure work, benefits, welfare protection, or representation, already generating significant risks and vulnerabilities in regular times that are dramatically exacerbated during

the COVID-19 pandemic. The uncertainty that transpires from this reality is further motivation to monitor and characterize how the lockdown measures were followed in the country to better understand the impact of the pandemic on the different portions of the population and help inform the response of public health and government officials.

L40 – This sentence seems like it should be included in the previous paragraph, and as written doesn't realty make sense. I would re-write it as: "For instance, internet adoption in rural areas of Mexico is estimated at 39.2% in urban areas with 53.8% of the population owning smartphones. In contract, 71.2% of urban residents have access to the internet and 77.7% use smartphones."

>Changed as recommended.

L41 – Bringing up DAS technology to monitor ground motions seems totally out of place here (seismic analysis hasn't even been introduced yet). This would be better to bring into the discussion.

>We moved the comment about DAS toward the end of the discussion when introducing the prevalence of traffic as a source of anthropogenic noise. The section reads as follows: "Anthropogenic seismic noise recorded in cities is typically generated by various processes such as our cultural and sporting events, individual and public commuter traffic and the production as well as transportation of different kinds of goods (e.g., Groos & Ritter, 2009). Traffic especially heavily influences the frequency range from âĹij1 Hz to more than 45 Hz with variation with respect to urban location due to the influence of subsurface conditions. As a result, geophysical sensing such as fiber‐optic distributed acoustic sensing (DAS) has recently proved capable to provide remarkably resolved statistics about public infrastructure utilization across many large sectors of a city (Lindsey et al., 2020)."

L49 – Another sentence that makes no sense as written. Suggest splitting into 2 sentences Suggested wording ": : :...correlates with mobility data (references). Therefore, analysis of seismic noise may off the ability to attain mobility information without the

privacyrelated concerns and limitations of using mobile technology."

>Changed as recommended.

L52: "Even though this area has lower seismicity rates than other parts of Mexico, it is..."

>Changed as recommended.

Table 1: Dates and events for "phase 1" are confusing because 2 dates are given. Can these be separate rows in the table or is the 14-03-2020 event needed?

>Both dates are associated with the phase 1, the distinction is referenced in the title of the column "Announcement/implementation dates". Authorities announced the phase 1 on 14-03-2020 to be implemented on 21-03-2020. However, schools and universities decided to immediately implement the recommended measures, and their impact on the anthropogenic noise is progressively observed before the official implementation date, hence the importance of emphasizing and associating both dates.

Table 2: Red row: suggest "encouraged" for "invited" – "invited to shelter at home" isn't really correct English.

>Corrected.

Figure 1 – Suggest a box outlining the "historic center" as stated on L78

>Figure updated as suggested.

L75 – Anthony et al, did not use Raspberry Shakes to densify networks as this sentence suggests - rather they tested the performance of the sensors and found them to be acceptable for use u regional networks. "from all magnitudes" is misleading – for instance, a Raspberry Shake cannot see a M5 teleseism. Finally, it might be nice to note that only data from the geophones is considered. Suggested edit: "These low-cost sensors were demonstrated to perform suitably well for monitoring a large range of local and regional earthquake magnitudes (Anthony et al., 2019). However, while

the accelerometers are capable or recording larger, more powerful earthquakes, they have high self-noise levels and are not able to resolve most cultural activity. Therefore, we restrict our analysis in this study to only the vertical component geophone data."

>We accept the suggestion and modified the section as follows: "These low-cost sensors were demonstrated to perform suitably well for monitoring a large range of local and regional earthquake magnitudes (Anthony et al., 2019). Although the accelerometers are capable of recording larger, more powerful earthquakes, they have high self-noise levels and are not able to resolve most cultural activity. Our analysis in this study is therefore restricted to the vertical component geophone data."

L81: Are these multiple universities? If so, it would be good to specify which Universities are included here.

>Specified as suggested: "The stations located further from the center of the city are installed in university campi (R2013 – UNAM Campus Juriquilla, R45B4 - Universidad Tecnológica de Querétaro, and R3F94 - Universidad Politécnica de Querétaro), ..."

L87 – remove "numerical conversion" – I'm not aware of this noise source unless it is digitizer noise, which I would think would fall under the "instrumental noise" umbrella.

>Changed as suggested.

L87 – Just leave it as "microseisms" – these are called "primary" and secondary microseisms (not first and second), but you won't be able to observe the primary with a Raspberry Shake.

>Changed as suggested.

L93 – Would probably be good to specify the amount of smoothing used in the Lecocq et al processing algorithm, especially since the data is then bandbassed into 1 Hz bins, and depending on smoothing in this step, these frequencies may not actually be being isolated.

>The smoothing used in the Lecocq et al. processing is reduced compared to the default IRIS MUSTANG and PQLX parameters to obtain a finer frequency resolution, in line with recommendations in Anthony et al. (2020, SRL). The smoothing is of 1/40 of an octave (as opposed to 1 for MUSTANG and PQLX) and the binning is of 1/80 of an octave (as opposed to ⅛ of an octave for MUSTANG and PQLX).

L94 – Would be good to make it clear that these are displacement RMS timeseries data (As opposed to acceleration or velocity).

>We modified the sentence to: "The time series were extracted from the root mean square (seismic RMS) of the time-domain displacement."

L95 – Would be good to specify how the 1 Hz RMS displacement bins were constructed. Was the data simply bandpassed? How many poles on the bandpass filter and where were they?

>The narrow 1 Hz bins were produced using a bandpass 4th order Butterworth filter between integer frequencies from 1 Hz to 20 Hz.

Figure 2 – Nice Figure! I would specify in the caption that what is being plotted is something along the lines of "normalized RMS displacement in 1 Hz frequency bins." I also suggest noting the move of R2013 directly in the Figure since the change of noise is so striking.

>Changed as suggested:

Figure 3 – It would also be good to specify in the caption the time-period over which the correlation analysis was performed as well as the meaning of the dashed lines (I assume they indicate the threshold for correlations to be significant at the 95% Confidence Interval?)

>We added in the caption that period considered is from 15 February to 1 October. The dashed lines were just references for -0.5 and 0.5 coefficients. The figure is now updated with a regular horizontal grid instead.

L116 – Can the normalization time period of the mobility data be changed to match that of the seismic (L138)?

>It unfortunately cannot be done without the raw community mobility data. Google only provides the normalized times series using the same method for all the available locations and warns of potential local and regional inadequacies.

L135 - Probably worth noting in the text why station R6BB7 was chosen for this analysis as well as that the determined 4-14 Hz frequency band is identical to that used in the Lecocq et al. (2020) study.

>We modified the sentence as follows: "Higher correlation coefficients between the community mobility and the Seismic RMS were obtained between 4 and 14 Hz, leading us to concentrate on that frequency range which is consistent with Lecocq et al. (2020)."

L152 – Is it possible that the work in the industrial park is seasonal? E.g. there is more activity in the spring and summer? It might be worthwhile to look at another year of data to make sure that seasonal trends are not being interpreted as COVID19-related here.

>The industrial park does not have any reported seasonal activity. It is unfortunately impossible to compare observations with previous years as the station was only installed in October 2019.

Figure 4 I don't think panels b-d were discussed in the text. Also, it wasn't clear to me if only a single day of data was being plotted or a date range. I suggest removing them from the manuscript if they are not needed. Figure 4A stands nicely on its own I think. . .

>Changed as suggested

It would be good to specify in the caption that this is 4-14 Hz noise (at least I think it is). What do the red and green shaded areas below the RMS noise curves mean?

[Figure]

>Changed as suggested. The red and green filling are visual aids to identify when the noise is respectively over or under the baseline level for the station.

The "blue" holiday lines look "green" to me in the Figure. It might be helpful to plot all stations on the same Y-axis to facilitate direct comparison between stations. It took me a while to realize the axis was not consistent.

>Updated as recommended.

L160 – Date of Easter would be helpful

>Added "(12 April 2020)" as the end of the Easter Holiday

L161- "..displayed the smallest change with noise levels dropping by less than 10%,.."

>Change as suggested

Figure 5 – Consider adding a dashed "0-line" to the plot as in Figure 4a. The holiday dashed lines look more black than blue to me: : :especially early on. It might be better to use solid and dashed lines here to distinguish between lockdown measures and holidays. Give date range of cross plots in Figures 5b and 5c. It wasn't clear to me over which time period these correlations were performed – particularly for Figure 5b

>Updated as suggested.

L177 – Suggest giving the data of "Good Friday Holiday"

>Added the "10 April 2020" as suggested.

L181 – Suggest referencing figure 4a here

>Added as suggested.

L190 – citations of Vidale (2011; SRL) and Diaz et al (2017, Scientific Reports) are probably warranted here as they also considered the seismic signatures of fans responding to scoring events at football games (Vidale is American football, but same idea).

>Added as suggested. The sentence now reads: "Seismometers in soccer stadiums are regularly used in Mexico as an entertaining tool to teach basic scientific research methodologies and seismology to undergraduate students (e.g., Melgar & Pérez-Campos, 2011), which echoes similar initiatives at sporting events worldwide (e.g., Vidale 2011; Díaz et al. 2017)"

L204 – This conclusion that the uptick in noise after the game is related to the hometeam winning seems to conflect with Figure 7, where the same uptick is seen in 2019 even though Gallos Blancos lost the game: : :..Could this uptick not be due to people starting their cars and driving home after the game?

>This conclusion was addressing the general enthusiasm of all the supporters rather than exclusively from the hometeam, hence this interpretation also applies to the 2019 game lost by Gallos Blancos. Also, the time window intentionally includes only a few minutes after the end of the game and is unlikely to include a lot of supporters driving out the stadium as it would be a rather progressive process as opposed to the sharp increase observed here (e.g., Boese et al., 2015, BSSA).

Furthermore, although Gallos is the home team, América is one of the most popular teams in the country and was likely to have a matching number of supporters, if not more than Gallos in the stadium. It should also be noted that in Mexico, a team losing a game is unlikely to translate into their supporters staying quiet and silent upon conclusion of the game.

Figures 6 and 7 - Label colorbar - Specify the frequency band plotted in the "E" panels. - Add median noise level for the week (as discussed in L205) to panel E

>Corrected

L232 – Over what time period were these correlation coefficients attained from? They don't match Figures 5b or 5c... I assume it is from a longer period than either of these plots?

[Figure]

>This was a mistake, coefficients in text included the pre-lockdown period. We updated them to only include the lockdown period, matching Figure 5C.

L234 – This sentence seems to argue that the only reason a positive correlation between seismic noise and people staying at home isn't observed is because the seismometers were not installed near residential areas. I don't think this is correct. I think the negative correlation with "residential areas" is expected as people don't make many seismic signals when they are sitting at home watching TV or working remotely. As is alluded to in the conclusion – most high frequency noise is likely coming from traffic – and this is simply reduced when people are staying at home.

>This is correct. This sentence attempted to swiftly discard a scenario in which the vicinity of a station is strictly residential. All stations in our network are hundreds of meters to a few kilometers of major commuting routes with traffic that generates a significant amount of anthropogenic seismic noise in normal times. In such a case, a strictly residential zone almost exclusively generates the typical outgoing and incoming traffic of the inhabitant commuters. With the shutdown of non-essential activities, major commuting routes observed a substantial decrease in traffic, due to people sheltering at home, which translated as the decrease of high-frequency seismic noise we observed. In a strictly residential zone, the commuter outgoing and incoming traffic is equally affected, but also replaced by traffic to and from essential activities/public places (e.g. increased deliveries, grocery shopping, parks, pharmacies, etc.). As a result, although the "residential mobility index" observed an increase during strict lockdown due to people spending more time home, a strictly residential area could see an increase in traffic generated by mechanisms other than the regular commuting of its inhabitants. In that specific scenario, a strictly residential area could see a decrease in traffic generated seismic noise less significant than the rest of the city.

Nevertheless, we acknowledge the potentially confusing nature of this explanation and decided to remove it for better clarity.

L242 – It seems like the argument is being made here that traffic noise is the dominant high-frequency noise source that is being modulated throughout the study? Can this be made more clear? Can you find some studies that show this? This is stated as a conclusion on L264, so a bit more support and clear thought that traffic is indeed the dominant noise source is warranted here

>The argument that traffic is the dominant source of anthropogenic noise in the selected frequency band is based on the scientific consensus on seismic noise in busy urban environments and the analysis of the configuration of our stations near major roads. This is a point that we make in the discussion L250-262.

To provide more support and clarity, we modified the final paragraph of the discussion as follows: "The seismic noise recorded by stations all within 5 km of the main roads in the state, a busy urban environment, is relatively similar across all stations in our frequency band of interest and has a specific temporal pattern that suggests a ubiquitous source. This has been characterized as a traffic-dominated anthropogenic noise (e.g., Boese et al., 2015; Green et al., 2017), and indicates that the method likely provides a good proxy for the temporal variation of traffic volume. With a lot of people working from home or losing their jobs during the lockdown, our results could in large part indirectly show the decrease in commuter traffic."

We also rewrote the following sentence in the conclusion to better carry this point across: "While we observed the contribution to anthropogenic noise in the near-field of source such as construction work and sporting events, the observed correlation between seismic noise and mobility at the city scale is consistent with the traffic-dominated nature of anthropogenic seismic noise in urban areas."

Please also note the supplement to this comment:
https://se.copernicus.org/preprints/se-2020-194/se-2020-194-AC1-supplement.pdf
* * *
Fig. 1.

Historic Center

R2013
R45B4
RB158
RF6B5
R6BB7
REBDD
R3F94

0 1 2 3 4 5 km

N

[Figure]

Fig. 2.

[Figure]

**Fig. 3.**

[Figure]

**Fig. 4.**

[Figure]

Fig. 5.

[Figure]

**Fig. 6.**

**Fig. 7.**

[Figure]

---

## Author Comment (AC2) · 14 Jan 2021

Review of SE_2020_194: Plaen et al., "Seismic signature... Querétaro, Mexico" Several papers have now been published describing the reduction of seismic ambient noise due to Covid-19 lockdown measures. So, this is not really an entirely new contribution. However, high frequency ambient seismic noise recorded in cities has many different contributing sources, mainly related to traffic, and it is not always easy to separate their sources and estimate their distance range. In this respect, the Querétaro case study, describing different noise behaviors in different parts of the town, is interesting and

deserves publication as it may contribute to further more detailed analyses. The paper is generally well written and clear, and the figures are of good quality. Some points, however, would benefit from better descriptions. I recommend publication with minor revisions, especially addressing the main points below.

>We are grateful for the positive comments and helpful recommendations.

Main points: 1) Mobility measures. It would be good to describe more clearly the meaning of the "Google mobility index". For example, does a 30% increase in residential mobility mean that there are 30% more people (i.e., 30% more cell phones) in residential areas, compared to the baseline period? Does the Google "mobility index" record only when people change their location ("movement trend" as in lines 113-114)? Or when they use their location app in the cell phones, even if not moving ("the time spent" in each category of place, as in lines 115-116). This is important to interpret the correlations of Fig. 5, for example.

>The confusion may arise from a difference in how the index is calculated for each category. The variation of the mobility index in the residential category is associated with the average amount of time in hours spent at places of residence by individual users relative to the count of unique users who spent any time at residences in a given day and geographic area. This method is specific to the residential category, as the workplace index is calculated as a count of users spending more than 1 hour at their places of work each day and is aggregated by place of residence of the users. In contrast, the other categories count the number of unique users who visited a public place of a given category in a given day. In this latter case, for each geographical location, individual users can contribute at most once to each category and up to 4 pairs of category-location per day (Aktay et al. 2020). We amended the explanation on the index for clarity.

2) In line 143, the sentence "The resulting time series show less impact from weekends, but they are characterized by lower noise levels ( Figure 4)" was not too clear for me.

[Figure]

You mean that noise on weekends also decreased compared to the baseline, but the decrease was not as large as for weekdays?

>Not exactly. The noise level is typically lower during weekends than during weekdays, this is the pattern that characterises the vast majority of urban environments. This pattern can still be observed with lockdown measures, probably due to essential activities mainly taking place during weekdays. However, when observed relative to the baseline (based on past corresponding days of the week), weekends exhibit a decrease that is consistent with the weekdays.

3) Lines 151-152 explains the increase in noise levels in the industrial area as possibly due to increase of "delivery of food and supplies". Is this just a hypothesis? Is there a way to help confirm this explanation?

>This is indeed a hypothesis. To the best of our knowledge the information necessary to confirm this explanation is not openly available and we had to perform an informed guess based on the possible essential activities that could generate such change.

4) section 3.2 Correlation of noise with mobility: 4.1) Lines 235-236. I did not quite understand the explanation for the anti-correlation with mobility in the residential areas. if increased mobility in the residential category indicates more people are staying at home, then less traffic will occur both in the residential areas as well as elsewhere in the city.

>As answered to Reviewer 1, this sentence attempted to swiftly discard a specific scenario in which a strictly residential area could see a decrease in traffic generated seismic noise less significant than the rest of the city. We acknowledge the potentially confusing nature of this explanation and removed it for better clarity.

4.2) Lines 236-239. I did not understand why the lower correlations in the "New norm" period, compared with the "Early lockdown phase" imply that the mobility pattern is "more complex" during the relaxation period. For example, a 30% mobility reduction

(assuming this is mainly related to traffic reduction) should produce the same RMS noise reduction, independent of the lockdown or new-norm period. I do not understand why traffic (the main source of seismic noise) should relate to the google mobility in different ways in the lockdown and in the new-norm phases. The problem, I think, is that correlation coefficients may not the best measure as they are much influenced by the scatter. The slope of the straight-lines in Fig. 5 may be a better proxy for the relationship between mobility and seismic noise. For example, the red lines (transit stations) have very similar slope in both phases, during the lockdown (Fig. 5b) and during the whole period (Fig. 5c): about 37% noise reduction/100% mobility reduction.

>The assertion that the mobility pattern is more complex during the Nueva Normali-dad than during the early lockdown phases is also associated with the limitations of Google's community mobility index. The location history is based on users' mobility regardless of whether they are driving or not. As a result, more uncertainty is intro-duced when analysing the relationship between the seismic noise and the mobility in-dex under the assumption that traffic is the dominating source of anthropogenic noise. With non-essential activities shut down during the early lockdown, fewer activities were available. As a result, the categories of mobility were likely more distinct and as their reduction translated into a decrease in traffic, the relationship with the decreasing an-thropogenic noise strengthened. When the city progressively re-opened, it triggered a progressive increase in mobility in all the non-residential categories, potentially more that 4 pairs of category-location per day (as limited by the Google algorithm) and not necessarily involving automotive transport. This increasingly complex mobility pattern has the potential to decrease the relationship between seismic noise and mobility as an increasing part of it (e.g. pedestrians and bicycles) does not generate traffic noise, the dominant source of anthropogenic seismic noise.

5) In section 4 (Conclusions) the authors seem to conclude that "traffic appeared to dominate observations" of seismic noise. Strictly speaking, the paper does not "prove" this link between traffic and seismic noise. However, all stations are near roads and

highways with heavy traffic ("< 5 km"); in addition, it is well known from the literature that traffic is the main contributor to seismic noise in stations close or within cities. So, it would be better to conclude that the observed correlation between seismic noise and mobility is consistent with the traffic-dominated nature of seismic noise in urban areas.

>The scope of the paper is not to prove that traffic is the dominant source of anthropogenic noise. For this purpose, we are limited in this study by the instruments available since the network was not designed for this purpose. As Reviewer 2 understood, the assertion that traffic dominates the noise in our observations is an interpretation based on the scientific consensus and the analysis of the configuration of the network near major roads. The point the paper is making is that although traffic typically dominates anthropogenic noise in cities and can be used to monitor mobility, sources in the near field such as road constructions, deliveries or event sporting/cultural events can significantly influence observations (e.g. R45B4, RB158 and REBDD). Although it may be tempting to use individual seismic stations to interpret the mobility of a whole city, we illustrate the limitations of such an approach. However, the average noise observed by a network at the city-scale offers a more reliable proxy for the variation of mobility, even with low-cost sensors.

We followed the suggestion and rewrote the following sentence to better carry this point across: "While we observed the contribution to anthropogenic noise in the near-field of source such as construction work and sporting events, the observed correlation between seismic noise and mobility at the city scale is consistent with the traffic-dominated nature of anthropogenic seismic noise in urban areas."

Minor points: a) Fig. 1: Please edit the position of label R6BB7 as it hides one of the other stations.

>Figure updated as suggested.

b) Fig. 4: Please indicate the "baseline period" (perhaps along the time axis) as it may help the readers understand the plot more easily.

>We updated the figure as recommended.

c) Fig. 5: I suggest to remove the probability level, so as not to clutter the figure, leaving only the r-value. This will also avoid hiding some of the data points. All probabilities are extremely small (because the number of points is very large) - it would be sufficient to mention in the text that all correlation coefficients are highly significant (probabilities less than 10E-15).

>Updated as suggested

d) Section 3.1 Sportive events: The analyses of the two football matches is interesting, but it does not add much. It only says that the noise from 22 players running in the field is insignificant compared with that of 30000 supporters jumping in the stadium.

>More than the lack of noise observed from the 22 players, the point is more the contrast with the noise observed with an audience and the perspective it offers when using a single station to characterise the mobility of a whole city. Beyond the entertainment factor of this section, it further emphasizes the benefit of using a network to study mobility at a city-scale, even using low-cost sensors.

e) Paragraph 230: the correlation coefficients mentioned in this paragraph are different from the ones in Fig. 5. Please check.

>Yes, this was a mistake. Those values corresponded to a past window selection that included time before the lockdown implementation. They are now updated to only include the lockdown period, matching Figure 5C.

Please also note the supplement to this comment:
https://se.copernicus.org/preprints/se-2020-194/se-2020-194-AC2-supplement.pdf

R2013

R45B4

RB158

RF6B5

R6BB7

REBDD

**Historic Center**

N

0 1 2 3 4 5 km

R3F94

**Fig. 1.**

[Figure]

**Fig. 2.**

[Figure]

Fig. 3.